# Clinical and Pathological Risk Factors for Peritoneal Metastases in a Surgical Series of T4 Colorectal Cancers

**DOI:** 10.3390/cancers17071103

**Published:** 2025-03-25

**Authors:** Dario Baratti, Carlo Galdino Riva, Marcello Guaglio, Tommaso Cavalleri, Gaia Colletti, Shigeki Kusamura, Giovanna Sabella, Massimo Milione, Elisabetta Kuhn, Francesca Laura Nava, Marcello Deraco

**Affiliations:** 1Peritoneal Malignancy Program, Department of Surgery, Fondazione IRCCS Istituto Nazionale dei Tumori, Via Venezian, 1, 20133 Milan, Italy; carlogaldino.riva@unimi.it (C.G.R.); marcello.guaglio@istitutotumori.mi.it (M.G.); tommaso.cavalleri@istitutotumori.mi.it (T.C.); gaia-colletti@istitutotumori.mi.it (G.C.); shigeki.kusamura@istitutotumori.mi.it (S.K.); francescalaura.nava@istitutotumori.mi.it (F.L.N.); marcello.deraco@istitutotumori.mi.it (M.D.); 21st Pathology Division, Department of Pathology and Laboratory Medicine, Fondazione IRCCS Istituto Nazionale dei Tumori, Via Venezian, 1, 20133 Milan, Italy; giovanna.sabella@istitutotumori.mi.it (G.S.); massimo.milione@istitutotumori.mi.it (M.M.); 3Department of Biomedical, Surgical and Dental Sciences, University of Milan, 20122 Milan, Italy; elisabetta.kuhn@unimi.it; 4Pathology Unit, Foundation IRCCS Ca’ Granda Ospedale Maggiore Policlinico, 20122 Milan, Italy

**Keywords:** colorectal cancer, peritoneal metastases, hyperthermic intraperitoneal chemotherapy

## Abstract

The interest in T4 colorectal cancer (CRC) has recently grown since different strategies based on the use of hyperthermic intraperitoneal chemotherapy (HIPEC) for the prevention or early treatment of peritoneal metastases have been clinically tested. However, the magnitude of the risk of peritoneal metastases in T4 CRC patients is not well defined. Analogously, the clinical/pathological factors that may help in identifying a subset of patients who deserve to be treated or not are still poorly understood. In our large surgical series, the prevalence of synchronous peritoneal metastases was 20.7%, and the 3-year cumulative incidence of metachronous peritoneal metastases was 21.5%. Negative lymph nodes and normal CEA-defined T4 tumours were at minimal risk for metachronous peritoneal dissemination. A potential implication of our observation is the possibility of more efficiently selecting higher-risk patients for the early detection and treatment of PM after curative resection, thus preventing overtreatment with an unfavourable harm/benefit ratio.

## 1. Introduction

Colorectal cancer (CRC) is the third most common tumour worldwide [1]. Peritoneal metastases (PM) from CRC are a primary cause of cancer-related morbidity and mortality. CRC-PM are commonly diagnosed at an advanced stage because small-volume disease is often asymptomatic and difficult to detect using current imaging studies [2,3,4,5,6,7,8]. The combination of cytoreductive surgery and hyperthermic intraperitoneal chemotherapy (CRS/HIPEC) has reportedly resulted in improved survival, but this treatment approach is maximally effective and safe only in highly selected patients with limited peritoneal disease [8]. On these bases, our group and other groups investigated different strategies for the prevention or early treatment of CRC-PM involving HIPEC either at the same time as primary surgery, in the early postoperative setting, or at second-look surgery after the completion of adjuvant systemic chemotherapy (s-CT) [9,10,11,12].

Since CRC-PM management is evolving towards a proactive approach, the identification of patients at higher risk has become increasingly relevant. As the main mechanism of PM development is the transcoelomic spread of tumour cells gaining access to the peritoneal cavity, CRC either perforating the visceral peritoneum (T4a) or directly invading surrounding organs (T4b) appears to be an important risk factor. However, while the T4 category carries an increased risk for PM, its risk level is reportedly lower than that of perforated tumours or a history of resected synchronous ovarian or peritoneal metastases [13]. Furthermore, it is unclear whether factors associated with PM in the general CRC population, such as nodal involvement, right-sided primary tumours, and mucinous histology, also increase PM risk in those with T4 tumours [2,3,4,5,13,14].

We reviewed our surgical series of T4a/b CRC to assess the occurrence and clinico-pathological risk factors for synchronous or metachronous PM. The ultimate aim of our study was to provide information on selecting patients more efficiently for protocols of the prevention or early detection of CRC-PM.

## 2. Materials and Methods

This is a single-centre surgical series from a high-volume CRC unit in a tertiary referral institution. This study was performed according to the Strengthening the Reporting of Observational Studies in Epidemiology (STROBE) guidelines [15].

### 2.1. Patient Population

We searched our institutional prospective database to identify 352 consecutive patients who underwent primary resection for pathological (p) or post-neoadjuvant therapy (yp) for T4a/b CRC (any N; any M) from January 2012 to June 2021 and retrieved information for this retrospective analysis. Clinical (c) T4a/b tumours were excluded, unless they were categorized as ypT4a/b after a pathological examination of the surgical specimens, owing to the difficulty of retrospectively discriminating between responses to preoperative treatments or preoperative staging inaccuracy.

All patients underwent an intensive preoperative work-up including physical examination; colonoscopy; contrast-enhanced thoracic, abdominal, and pelvic computed tomography (CT); and CEA and CA19.9 measurements. Patients with rectal tumours had pelvic nuclear resonance imaging and endoscopic ultrasound. We performed additional studies, such as fluorodeoxyglucose positron emission tomography, as needed. All patients were discussed in multidisciplinary meetings.

### 2.2. Operative Treatment

Our operative techniques were previously described [16,17]. Up until November 2019, patients routinely underwent open surgery. After that date, videolaparoscopic surgery (VLS) was increasingly used, according to an institutional program for VLS implementation [16]. Primary resections were performed according to the oncologic principles of adequate lymphadenectomy with central vascular ligation. Total mesorectal excision was performed in rectal tumours, and a distal margin ≥2 cm was ensured for upper third rectal cancers, along with appropriate circumferential resection margins. Preoperative radiotherapy for locally advanced tumours entirely or partially located below the peritoneal reflection was given at a dose of 45/50 Gy in 25 days, with concurrent oral capecitabine, or as a short-term course (25 Gy in 5 consecutive days).

Postoperative complications occurring within 30 days were scored according to the National Cancer Institute Common Terminology Criteria for Adverse Events, version 4.0 [18]. Postoperative adjuvant s-CT was administered according to international guidelines [19]. All patients underwent postoperative follow-up, consisting of a physical examination, a thoracic/abdominal CT scan, and marker measurements performed every three months during the first 2 years and every six months thereafter.

### 2.3. Pathological Assessment

All surgical specimens were evaluated by two expert gastro-intestinal pathologists during the study period. The International Union Against Cancer (UICC)/American Joint Committee on Cancer (AJCC) TNM classification, 8th edition, was used to retrospectively stage tumours operated on before 2018 [20]. The following features were recorded: T4a vs. T4b subcategory; number of involved and sampled regional lymph nodes; pathological grading; extent of mucinous and signet ring cell component; infiltrative vs. expansive pattern of invasiveness; presence of ulceration; Crohn’s-like lymphoid reaction component; lymphatic, perineural, intratumoral, and extratumoral vascular invasion; peritumoral and intratumoral infiltrating lymphocytes; and involvement of distal, proximal, and circumferential surgical margins. All *KRAS*, *BRAF*, and NRAS mutational analyses were performed for clinical purposes by Sanger sequencing, as described elsewhere [21]. Mismatch repair deficiency (dMMR) and high microsatellite instability (MSI-H) were identified by immunohistochemistry or the multiplex polymerase chain reaction system, respectively.

### 2.4. Definitions

As the present study focuses on peritoneal dissemination, we defined the primary site as follows:Right-sided tumours: lesions located proximally to the splenic flexure;Left-sided tumours: lesions located from the splenic flexure to the peritoneal reflection;Extraperitoneal tumours: lesions with the cranial limit below the peritoneal reflection, infiltrating surrounding structures, such as genitourinary organs, anal musculature, sacrum, pelvis sidewall, or floor (T4b).

As all of our patients underwent abdominal exploration, all peritoneal, abdominal wall, omental, and ovarian metastases diagnosed before or at primary resection were considered synchronous PM, and as for metachronous PM, we considered those diagnosed after primary resection. Disease relapses close to the initial tumour location, such as anastomotic, adjacent mesenteric, regional lymph node, and pelvic recurrences in rectal tumours, were defined as local recurrences. All recurrences that did not meet the conditions of PM or local recurrence, such as lung, liver, or non-regional node metastases, were defined as distant metastases.

### 2.5. Statistics and Study Design

This study’s flow-chart is shown in Figure 1. The prevalence of synchronous PM was calculated in the overall series. We assessed baseline differences between groups by using Student’s *t* test, a Chi-square test, or Fisher’s exact test, as appropriate. Baseline variables significantly associated with synchronous PM at univariate analysis were assessed by multivariate logistic regression models.

For the analysis of metachronous PM, this study ended on June 30, 2022; thus, all patients had a minimum 12-month potential follow-up. Cases of operative deaths, patients with synchronous metastases (all-type), or a follow-up <12 months were excluded. As our institution is a tertiary referral centre for the whole country, a number of patients returned to referring hospitals for postoperative therapies and follow-up. The cumulative incidence of metachronous PM was calculated, considering death as a competing event. The influence of clinico-pathological variables on the hazard of metachronous PM was assessed by Cox proportional hazard models using the backward elimination method.

Continuous variables were categorized into two classes by using their mean value as a cut-off. *p* values < 0.05 were considered significant. All statistical analyses were conducted by SPSS, version 20.0.0 for Windows (SPSS, Chicago, IL, USA).

## 3. Results

The 352 patients included in the present analysis accounted for 11.1% of the 3171 primary CRC resections performed during the study period. The main patient, tumour, and treatment characteristics are shown in Table 1, according to the presence of synchronous PM. Other pathological and biological features are shown in Appendix A. VLS was performed in 15 patients (4.3%), including two conversions to open surgery. Only four (1.1%) patients were operated on in an emergency setting. Severe (grades 3–5) postoperative complications occurred in 69 patients (19.6%), with three in-hospital deaths (0.9%).

### 3.1. Synchronous Peritoneal Metastases

In the overall series, the prevalence of all-type synchronous metastases was 145/352 (41.2%), and the prevalence of PM was 73/352 (20.7%). A total of 44 of 73 patients (60.3%) only had PM. The peritoneum was the second most commonly involved site in 109 patients with a single site of metastasis, after the liver (n = 49) and before the lungs (n = 8), distant nodes (n = 6), and bone (n = 2). Additionally, 37 patients had multiple organs affected (liver and lung, n = 6; peritoneum and liver, n = 24; peritoneum and lung, n = 1; peritoneum and distant nodes, n = 2; liver and distant nodes, n = 1; peritoneum, liver, and lung, n = 2). Therefore, the peritoneum was also the second most commonly involved site in patients with either one or multiple sites of metastasis, after the liver (n = 83) and before the lungs (n = 17), distant nodes (n = 9), and bone (n = 2).

Synchronous PM were found preoperatively in 43 patients (58.9%) and during surgery in 30 (41.1%). The treatment used was CRS/HIPEC in 17 patients, macroscopically complete CRS in 32, partial debulking in 10, palliative s-CT in 13, and best supportive care in 2.

### 3.2. Risk Factors for Synchronous Peritoneal Metastases

PM were observed in 1/58 (1.7%) tumours below the peritoneal reflection and 72/294 (24.5%) above the peritoneal reflection (colon or proximal rectum). The difference was highly significant (*p* < 0.001). Moreover, primary site, age ≤65.6 (*p* < 0.001), elevated CEA (*p* = 0.002), elevated CA19.9 (*p* < 0.001), no preoperative RT (*p* = 0.008), and extra-PM (*p* = 0.020) were correlated with synchronous PM. Among pathological features, mucinous/signet ring cell histology (*p* < 0.001), positive lymph nodes (*p* = 0.001), and absent/mild peritumoral (*p* = 0.001) and intratumoral (*p* = 0.001) infiltrating lymphocytes were associated with PM.

The results of a multivariable model including 345 patients with complete data are shown in Table 2: age ≤65.5 (*p* = 0.037), primary site (*p* = 0.002), positive nodes (*p* = 0.005), elevated CA19.9 (*p* = 0.001), and mucinous/signet ring cell histology (*p* = 0.001) retained their significance. Peritumoral and intratumoral infiltrating lymphocytes were excluded from this model and included in a second model collecting 265 patients with available data, together with the above listed variables. However, peritumoral (*p* = 0.200) and intratumoral infiltrating lymphocytes (*p* = 0.127) did not reach statistical significance (data not shown).

### 3.3. Metachronous Peritoneal Metastases

After the exclusion of operative deaths (n = 3), patients with synchronous metastases (n = 145), or follow-up <12 months (n = 40), an analysis of metachronous PM was performed in 164 patients. After a median reverse Kaplan–Maier follow-up of 35.9 months (95% confidence interval [CI] = 29.5–44.9), metachronous PM occurred in 36 patients (22.0%). The cumulative incidence of metachronous PM is shown in Figure 2. One-, two-, and three-year cumulative incidence were 11.5% (95% CI = 5.7–15.2), 17.7% (95% CI = 11.4–23.6), and 21.5% (95% CI = 14.3–28.1), respectively.

The median time to metachronous PM was 11.7 months (range 1.3–83.1). PM were diagnosed within 36 months in 31 patients (86.1%). Isolated PM occurred in 24 patients (66.6%) and were treated by CRS/HIPEC in 1 patient, macroscopically complete CRS in 9, palliative s-CT in 12, and BSC in 1. In 12 patients, metachronous PM were associated with extra-PM (33.3%) and were treated by macroscopically complete surgery in 2, palliative s-CT in 9, and BSC in 1.

### 3.4. Risk Factors for Metachronous Peritoneal Metastases

An analysis of factors correlated with the risk for metachronous PM is shown in Table 3. In univariate analysis, age > 65.5 (*p* = 0.033), right-sided primary tumour (*p* = 0.045), positive lymph nodes (*p* = 0.011), CEA > 5.0 (*p* = 0.021), positive margins (*p* = 0.008), and no postoperative s-CT (*p* = 0.035) were correlated with a higher risk of PM.

We tried to combine factors univariately associated with metachronous PM to identify subgroups at different risk levels. The administration of postoperative s-CT was excluded because it had no value in the early postoperative risk assessment process. The combination of CEA level and nodal status offered the best discrimination. Metachronous PM occurred in 3 of 48 patients with negative nodes and normal CEA (3-year cumulative incidence 4.6%; 95%CI = 1.2–17.8%), as compared with 10 of 38 of those with negative nodes and elevated CEA (3-year cumulative incidence 27.1%; 95%CI = 15.4–47.9%), 12 of 42 of those with positive nodes and normal CEA (3-year cumulative incidence 31.5%; 95%CI = 18.4–54.1%), and 11 of 36 of those with positive nodes and elevated CEA (3-year cumulative incidence 31.0%; 95%CI = 18.3–52.6%) (see Figure 3).

After multivariate analysis, only combined nodal status and preoperative CEA (*p* = 0.033), no postoperative s-CT (*p* = 0.001), positive margins (*p* = 0.008), and, with borderline significance, age (*p* = 0.052) were independently associated with PM.

## 4. Discussion

The main objectives of the present study were to quantify the risk for PM in T4a/b colorectal cancer and investigate clinical/pathological risk factors for PM development. The management of these patients continues to evolve, as promising results have been obtained from randomized trials of total neoadjuvant chemotherapy in rectal cancer and 6-week preoperative s-CT in radiological T3–4 operable colon cancer [22,23]. Nevertheless, the eradication of microscopic peritoneal seeding in patients undergoing curative-intent surgery remains a major unmet clinical need. To our knowledge, no studies have demonstrated that adjuvant s-CT reduces the rate of peritoneal relapse in pT4 CRC [12], and the FOxTROT trial provided no information on the impact of neoadjuvant s-CT on peritoneal disease control [23]. Understanding potential differences in oncological outcomes within this heterogeneous and high-risk subgroup of CRC is instrumental in designing tailored strategies to diagnose and treat PM at an early stage or prevent their development [8,9,10,11,12,13,14].

The magnitude of the risk for PM in T4 CRC is still poorly defined. In recent studies, rates range widely from 9.9% to 23.0% for synchronous PM and from 10.1% to 42.1% for metachronous PM, depending on the inclusion of patients with rectal tumours, emergency surgery, and distant metastases (see Table 4) [2,3,4,5,7,10,12,24,25,26,27,28,29,30,31,32]. We chose to include every presentation (both elective and emergency) and any site of disease to better reflect our daily practice. In our large surgical series, synchronous PM were diagnosed in approximately 20% of patients and metachronous PM in 22% of the 164 evaluable patients, in line with the existing literature.

A small number of studies have addressed the risk factors for PM in T4 CRC. van Sandvoort and Klaver did not find any correlation [25,26]. Bastianen reported female sex, right-sided primaries, peritumoral abscess, pT4a and pN2 categories, R1 resection, signet ring cell histology, and postoperative infections as independent predictors to conclude that pT4a vs. pT4b is the main determinant of outcome [29]. Finally, Tsai and Cerdan-Santacuz developed prediction models to categorize patients into different risk groups, but these models are not yet widely used in clinical practice [28,32].

The interest in T4 CRC has recently grown since a Spanish randomized trial demonstrated an increase in peritoneal disease control associated with mitomycin-based HIPEC performed simultaneously with primary surgery [12]. This successful trial follows two failed randomized studies testing different strategies [10,11]. Oxaliplatin-based HIPEC was delivered five–eight weeks after surgery for pT4 or perforated primary tumours in the COLOPEC trial to prevent peritoneal recurrence [10] and at second-look surgery after the resection of CRC with perforation and/or ovarian or peritoneal metastasese in the PROPHYLOCHIP trial to diagnose and treat early-stage disease. Unfortunately, substantial differences in study design make it difficult to define the optimal drug, timing, and selection criteria for the proactive management of high-risk patients.

An important finding of our study is the identification of a subset of patients at minimal risk for metachronous PM, defined by pathologically negative lymph nodes and normal CEA. This group accounted for 29.3% of 164 patients with no synchronous peritoneal or distant metastases who can potentially be cured by surgery and adjuvant therapies. A potential implication of our observation is the possibility of more efficiently selecting higher-risk patients for the early detection and treatment of PM after curative resection, thus preventing overtreatment with an unfavourable harm/benefit ratio. Furthermore, these combined variables may be used to better stratify the risk of developing metachronous PM in future trials.

The HIPECT4 trial results and a theoretical issue (i.e., the possibility of preventing tumour cell entrapment in postoperative adhesions) would suggest that adjuvant HIPEC should ideally be administered at the time of primary resection [12]. Accordingly, node status, which can be determined only by pathological examination after surgery, would be of little help in patient selection. Nevertheless, the current evidence does not definitively support the superiority of the simultaneous time setting, as the clinical benefits seen in the Spanish trial could be related to the use of mitomycin instead of oxaliplatin, rather than to the simultaneous timing, instead of delayed, to deliver adjuvant HIPEC.

Furthermore, the clinical diagnosis of T4 CRC is still challenging. In the HIPECT4 trial, only 125 of 184 cT4 colon cancers (67.9%) were confirmed as pT4 at pathological examination [12]. Similarly, the pT4 category was not identified preoperatively in 92 of 200 patients (54%) in a Dutch surgical series [25] and in 61% of patients in a population-based study by Klaver et al. [33]. Since delivering prophylactic HIPEC for clinical T4 CRC simultaneously with primary surgery will inevitably miss unexpected pT4 tumours, other strategies should be developed for these patients. The COLOPEC-2 trial (NCT03413254) will clarify the impact of intensified surveillance in pT4 CRC. After primary treatment, patients will undergo a second look laparoscopy, and those without evidence of disease will be randomized to further follow-up with or without a third-look diagnostic laparoscopy at 18 months [34]. Other trials are focusing on adjuvant treatments, such as our study prospectively assessing adjuvant Pressurized Intraperitoneal Aerosolized Chemotherapy (PIPAC) in pT4a/b CRC within 4–8 weeks from primary surgery (NCT03413254) [35].

As compared with the studies outlined in Table 4, a strength of the present series is the larger number of pathological and biological features analyzed for a possible correlation with PM. Nevertheless, only peritumoral and intratumoral lymphocytic infiltrates were correlated with synchronous PM but not with metachronous PM. Lymphocytic reaction has been associated with improved survival [36], but the lack of significance in multivariate analysis suggests that the role of the host immune response deserves further investigation in this setting. Analogously, previous series of palliative s-CT have reported the association of CRC-PM with mutated BRAF, but such a correlation was not seen in our patients [21].

The 20.7% rate of synchronous PM in our study suggests that the radiological suspicion of a T4 tumour needs special attention for the diagnosis of PM during primary resection. We identified several variables associated with synchronous PM, but their clinical usefulness is inherently limited by their poor sensitivity to preoperatively recognizing the T4 category. Nevertheless, in colon tumours with areas of penetration into the peritoneum or other organs by radiological imaging, a preliminary laparoscopic exploration may be considered, especially when adjunctive risk factors such as younger age, non-intestinal histology, or elevated CA19.9 are present.

We acknowledge several limitations of the present analysis, including its retrospective design. Our highly selected study population could have limited the generalizability of the findings outside a high-volume tertiary centre. The small number of emergency presentations was due to the fact that this series is from a comprehensive cancer centre without an emergency department. Another limitation could be the interobserver variability in pathological assessments, since surgical specimens were not reviewed for the purpose of the present study. However, two expert pathologists (MM and GS) assessed all the cases during the whole study period. Furthermore, missing data regarding MMR status and RAS/RAF mutations could have resulted in failure to detect possible statistical correlations with PM occurrence. Finally, the assessment of metachronous PM was largely based on imaging studies, thus leading to the underestimation of their incidence.

## 5. Conclusions

The T4 category represents a clinically relevant risk factor for both synchronous and metachronous PM. The peritoneum is a common site of recurrence after potentially curative primary surgery, highlighting the need for peritoneal prophylactic treatments. Negative lymph nodes and normal preoperative CEA define a subset of patients at lower risk for metachronous PM within the T4 CRC population and may be used as selection factors in tailored protocols for the prevention and early treatment of PM.

## Figures and Tables

**Figure 1 cancers-17-01103-f001:**
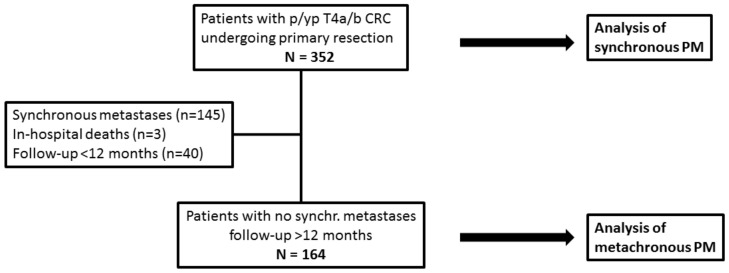
Study flow-chart. CRC: colorectal cancer; PM: peritoneal metastases; p: pathological; yp: post-neoadjuvant therapy.

**Figure 2 cancers-17-01103-f002:**
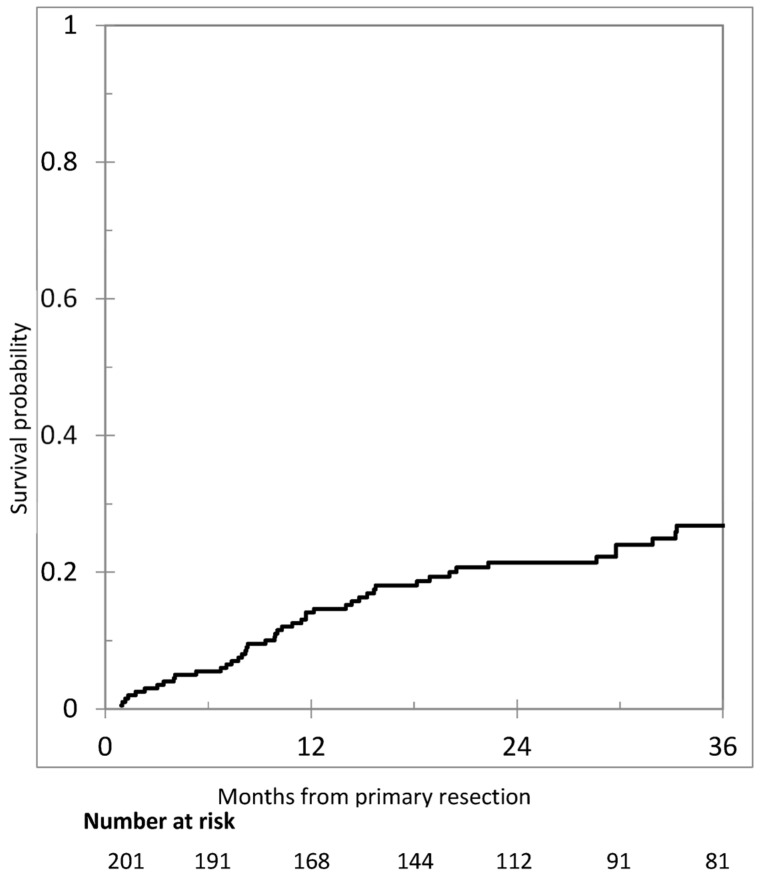
Cumulative incidence of metachronous peritoneal metastases in 201 patients.

**Figure 3 cancers-17-01103-f003:**
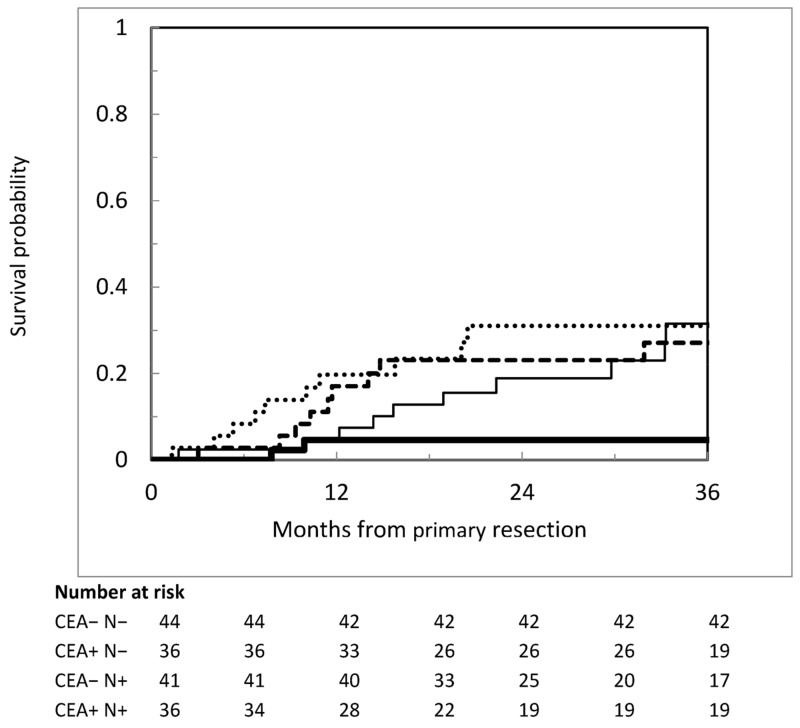
Cumulative incidence of metachronous peritoneal metastases according to pathological lymph node status and CEA level. Heavier continuous line: negative nodes and normal CEA; lighter continuous line: negative nodes and increased CEA; dashed darker line: positive nodes and normal CEA; dotted line: positive nodes and increased CEA.

**Table 1 cancers-17-01103-t001:** Patient characteristics according to the presence vs. absence of synchronous peritoneal metastases.

Variables	Categories	N.	%	Synchronous Peritoneal Metastases				
				yes N.	(n = 73) %	no N.	(n = 279) %	*p* Value
Sex	Male Female	178 174	50.6 49.4	36 37	20.2 21.3	142 137	79.8 78.7	0.896
Age	Median, IQ range	65.5 55.1–74.2		58.4	48.6–71.7	65.5	57.3–75.6	<0.001
Race	White Asian	349 3	99.1 0.9	73 0	20.9 -	276 3	79.1 100.0	1.000
Site of primary tumour	Right-sided Left-sided Below perit. refl.	111 183 58	31.5 52.0 16.5	31 41 1	27.9 22.4 1.7	80 142 57	72.1 77.6 98.3	<0.001
T category	p/yp 4a p/yp 4b	230 122	65.3 34.7	47 26	64.4 35.6	183 96	65.6 34.4	0.890
N category	0 1a/b/c 2a/b	123 123 102	35.3 35.3 29.4	13 27 31	10.6 22.0 30.4	110 96 71	89.4 78.0 69.6	<0.001
M category	0 1a 1b 1c	207 65 7 73	58.8 18.5 2.0 20.7	0 0 0 73	- - - 100.0	207 65 7 0	74.2 23.3 2.5 -	NA
Extraperitoneal disease	Present Absent	100 252	28.4 71.6	29 44	29.0 17.5	71 208	71.0 82.5	0.020
CEA	>5.0 ng/mL ≤5.0 ng/mL	189 159	54.3 45.7	51 21	35.6 13.2	138 138	64.4 86.8	0.002
CA19.9	>37.0 UI/mL ≤37.0 UI/mL	104 244	41.9 58.1	37 35	51.4 48.6	67 209	24.3 75.7	<0.001
Preoperative CT	Conducted Not conducted	86 266	24.4 75.6	23 50	26.7 18.8	63 216	73.3 81.2	0.127
Preoperative RT	Conducted Not conducted	61 291	17.3 82.7	5 68	8.2 23.4	56 223	91.8 76.6	0.008
Surgery	Right/transverse Left/sigmoid Total/subtotal RAR Hartmann APR Other	110 87 5 99 27 21 3	31.2 24.7 1.4 28.1 7.7 6.0 0.8	29 26 0 10 7 0 1	29.1 21.9 1.8 31.9 7.1 7.5 0.7	81 61 5 89 20 21 2	39.7 35.6 0 13.7 9.6 0 1.4	NA
Postoperative complications	Yes No	69 283	19.6 80.4	10 63	14.5 22.3	59 220	85.5 77.7	0.186
Postoperative CT	Conducted Not conducted	244 59	80.5 19.5	51 12	20.9 20.3	193 47	79.1 79.7	NA
Postoperative RT	Conducted Not conducted	29 323	8.2 91.8		- 22.6	29 250	100.0 77.4	NA

NA: not available/not assessed; IQ: interquartile; CT: chemotherapy; RT: radiotherapy; RAR: rectum anterior resection; APR: abdominal perineal resection; p: pathological; yp: post-therapy.

**Table 2 cancers-17-01103-t002:** Multivariate analysis of factors associated with synchronous peritoneal metastases.

Variables	Categories	OR	95%CI	*p* Value
Age	≤65.5 vs. >65.5	1.85	1.04–3.33	0.037
Site of primary tumour	Below perit. reflection vs. left-sided vs. right-sided	2.08	1.30–3.33	0.002
CEA	>5.0 ng/mL vs. ≤5.0 ng/mL	1.63	0.86–3.12	0.136
CA19.9	>37.0 UI/mL vs. ≤37.0 UI/mL	2.93	1.62–5.30	<0.001
Histological type	Intestinal vs. mucinous/SRC	3.08	1.58–5.98	<0.001
N category	2 vs. 1 vs. 0	1.70	1.18–2.46	0.005
Extraperitoneal disease	Absent vs. present	1.29	0.67–2.50	0.443
Preoperative RT	Conducted vs. not conducted	0.75	0.24–2.36	0.626

OR: odds ratio; CI: confidence interval; RT: radiotherapy; SRC: signet ring cell; N: node.

**Table 3 cancers-17-01103-t003:** Univariate and multivariate analysis of factors associated with metachronous peritoneal metastases in 201 patients with complete data.

Variables	Categories	Univariate			Multivariate		
		HR	95% CI	*p* Value	HR	95% CI	*p* Value
Sex	Male vs. female	0.94	0.70–1.27	0.696			
Age	≤65.5 vs. >65.5	1.39	1.03–1.88	0.033	1.37	0.98–1.89	0.052
Primary site	Below perit. reflection Right-sided Left-sided	Ref. 0.75 0.80	0.51–1.11 0.64–0.99	0.153 0.045	0.94	0.75–1.17	0.564
CEA	>5 ng/mL vs. ≤5 ng/mL	1.43	1.06–1.95	0.021			
CA19.9	>37 UI/mL vs. ≤37 UI/mL	1.33	0.91–1.94	0.139			
Histological type	Intest. vs. mucinous/other	1.22	0.78–1.90	0.382			
Grading	3 vs. 1/2	1.26	0.91–1.74	0.165			
T category	4a vs. 4b	0.73	0.52–1.02	0.066			
N category	2 vs. 1 vs. 0	1.31	1.08–1.59	0.006	1.27	1.02–1.59	0.033 *
Extraperit. disease	Yes vs. no	2.1	1.51–3.08	<0.001	2.07	1.40–3.06	<0.001
Ulceration	Yes vs. no	0.89	0.44–1.75	0.727			
Intratumoral vascular invasion	Yes vs. no	0.98	0.61–1.57	0.927			
Peritumoral vascular invasion	Yes vs. no	0.84	0.53–1.32	0.442			
Neural invasion	Yes vs. no	1.22	0.76–1.95	0.406			
Pattern of invasiveness	Infiltrative vs. expansive	0.82	0.54–1.22	0.324			
Peritumoral infiltrat. lymph.	Absent/mild vs. moderate/severe	0.85	0.61–1.20	0.366			
Intratumoral infiltrat. lymph.	Absent/mild vs. moderate/severe	1.12	0.79–1.60	0.528			
Crohn’s like lymphoid reaction	Yes vs. no	1.18	0.73–1.92	0.490			
Margins	R1/2 vs. R0	1.91	1.19–3.09	0.008	2.01	1.20–3.39	0.008
Preoperative CT	Conducted vs. not conducted	0.94	0.67–1.33	0.731			
Postoperative CT	Conducted vs. not conducted	0.67	0.46–0.97	0.035	0.51	0.33–0.77	0.001
Preoperative RT	Conducted vs. not conducted	0.88	0.62–1.26	0.494			
MSS	MSI-H/dMMR vs. MSS/pMMR	0.95	0.51–1.75	0.857			
KRAS	Mut. vs. wild-type	0.81	0.52–1.27	0.366			
NRAS	Mut. vs. wild-type	0.85	0.26–2.71	0.779			
BRAF	Mut. vs. wild-type	1.27	0.60–2.66	0.531			
Complications	Yes vs. no	1.13	0.77–1.66	0.518			

HR: hazard ratio; CI: confidence interval. CT: chemotherapy; RT: radiotherapy; R0: microscopically free; R1: microscopically involved; R2: macroscopically involved; MSS: microsatellite stability; MSI-H: high microsatellite instability; pMMR: mismatch repair proficiency; dMMR: mismatch repair deficiency. * refers to combined CEA and lymph node status.

**Table 4 cancers-17-01103-t004:** Selected series of peritoneal metastases in T4-stage colorectal cancer.

Author (Year)	Study Design	Site of Primary Tumour	N. of pts	PM	%	Timing
Lemmens 2011 [2]	Pop.-based	C + R	1986	324	16.2	Synchr.
Segelman 2012 [3]	Pop.-based	C + R	1138	281	24.7	Synchr. + metachr.
Hompes 2012 [24]	Surg. series	C	19	8	42.1	Metachr.
Kerscher 2013 [4]	Surg. series	C + R	378 306	72 48	19.0 15.7	Synchr. Metachr.
Van Gestel 2014 [5]	Pop.-based	C + R	487	9	10.1	Metachr.
Van Santvoort 2014 [25]	Surg. series	C + R	200 154 200	46 33 79	23.0 21.2 39.5	Synchr. Metachr. Synchr. + metachr.
Klaver 2018 [26]	Surg. series	C	159 130	29 30	18.2 23.1	Synchr. Metachr.
Klaver 2019 [10]	Rand. trial	C	100 102	19 23	19.0 22.5	Metachr. Metachr.
Arrizabalaga 2020 [27]	Surg. series	C	125 98	15 21	12.0 21.4	Synchr. Metachr.
Bastianen 2021 [29]	Surg. series	C	852	156	18.3	Metachr.
Tsai 2021 [28]	Surg. series	C	2003	246	12.3	Metachr.
Lurvink 2021 [7]	Pop.-based	C	192 119	19 19	9.9 16.0	Synchr. Metachr.
Uppal 2021 [31]	Surg. series	C	151	27	17.9	Metachr.
Li 2021 [30]	Surg. series	C	195	73	37.4	Metachr.
Arjona-Sánchez 2022 [12]	Rand. trial	C	62	10	16.1	Metachr.
Cerdán-Santacruz 2022 [32]	Surg. series	C	1356	185	13.6	Metachr.

PM: peritoneal metastases; C: colon; R: rectum.

## Data Availability

The data presented in this study are available on request from the corresponding author due to legal reasons.

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
