# Peer review of "Clinical and Pathological Risk Factors for Peritoneal Metastases in a Surgical Series of T4 Colorectal Cancers"

_cancers, 2025, doi:10.3390/cancers17071103_

Round 1
Reviewer 1 Report
Comments and Suggestions for Authors
This article is interesting because it investigated risk factor of peritoneal metastasis in colorectal cancer known as worst prognosis.
major comments:
1/the size of table one is too long. It will be more clear if only relevant and significant parameters will be present and put the other ones in a supplemental table. Also better organized them in clinical, epidemiological, biological subsections in the table 1
2/ Figure 3 is not readable: 4 classes or presented only three group at risk are described and for the two intermediary groups are in dash line so impossible to distinguish between them according the legend. The line of the worst group is not visible.
Author Response
Comment 1: the size of table one is too long. It will be more clear if only relevant and significant parameters will be present and put the other ones in a supplemental table. Also better organized them in clinical, epidemiological, biological subsections in the table 1.
Response 1: Thank you for pointing out. We agree with this comment. Therefore, we have changed Table 1 including only patient and treatment-related varibles according to the presence vs. absence of synchronous peritoneal metastases. Pathological and biological features are now displayed in Supplementary table. The second sentence of the Results section (page 5, lines 168 to 170) was modified accordingly.
Comment.2: Figure 3 is not readable: 4 classes or presented only three group at risk are described and for the two intermediary groups are in dash line so impossible to distinguish between them according the legend. The line of the worst group is not visible.
Response 2: We thank the reviewer for his/her helpful suggestions. Therefore, numbers of patients at risk were added for the fourth group of patients with both high CEA and positive lymph-nodes. Figure legend was corrected. We apologize for the typing error. In the revised version of the table, the lighter continue line represents patients with negative nodes and increased CEA, and the dashed darker linerepresent patients with positive nodes and normal CEA. We do believe that these changes make the table easier to read and understand. The line for the worst group (patients with both with both high CEA and positive lymph-nodes) is now visible in the lower part of the diagram. Figere 3 legend on page 11, lines 315-318 was modified.
Reviewer 2 Report
Comments and Suggestions for Authors
Please see attached word file.

Please see attached word file.
Author Response
We thank the reviewer for the helpful comments. Please, find our replays in the attached file.

Reviewer 3 Report
Comments and Suggestions for Authors
This is a very well written article which aims to quantify the risk for peritoneal metastases in T4a/b colorectal cancer and to investigate clinical-pathological risk factors for peritoneal mestastases development.
The topic is extremely important in the field, considering the huge number of cases that we all have to deal with in the last years.
The authors present here a prospective study on 352 patients undergoing T4 primary colorectal resection from 2012 to 2021.
The data are very well presented in the tables, making them very easy to read. The statistical part looks very good.
I would only have one mention about the editing part: shouldn't the points follow the square brackets? I observed in the entire text that they are put before the square brackets.
I think this is a good article, which deserves to be published in Cancers.
Author Response
We agree with this comment and thank the reviewer. We have carefully revised manuscript to have all the reference numbers btween square braketts before the punctuation, according to this journal guidelines.
Reviewer 4 Report
Comments and Suggestions for Authors
Thank you for allowing me to review this monocentric study evaluating the risk factors for the occurrence of synchronous and metachronous peritoneal metastases in the T4 colorectal cancer subgroup. This is a retrospective analysis using data from a proprietary database. the idea behind the paper is original and relevant in view of the poor prognosis and heterogeneity of T4 colorectal tumours. the manuscript is well written and illustrated.
I do, however, have a few comments and questions
Why exclude clinical T4 tumours?
Why did you include colorectal tumours complicating Crohn's disease, given that their clinical history is different?
The authors state that a distal margin of more than 2 cm must be respected in cancers of the upper rectum. To my knowledge, the exeresis must extend beyond 4 cm because of the risk of tumour islands in the mesorectum, which are present in up to 80% of cases.
what classification was used to define post-operative morbidity?
what time frame was used to define the metachronous nature of peritoneal metastases?
Of the 73 patients with synchronous peritoneal metastases, why was neoadjuvant chemotherapy only given to 23 of them?
What was the post-operative morbidity? Was it assessed in the statistical analysis of risk factors for synchronous peritoneal metastases?
Finally, in the discussion, I will look at the role of neo-adjuvant treatment in T4
Author Response
Thank you for allowing me to review this monocentric study evaluating the risk factors for the occurrence of synchronous and metachronous peritoneal metastases in the T4 colorectal cancer subgroup. This is a retrospective analysis using data from a proprietary database. The idea behind the paper is original and relevant in view of the poor prognosis and heterogeneity of T4 colorectal tumours. The manuscript is well written and illustrated.
AUTHORS’ RESPONSE We thank the reviewer for the encouraging comments.
I do, however, have a few comments and questions
Why exclude clinical T4 tumours?
AUTHORS’ RESPONSE: clinical T4a/b tumours that were staged as ypT1-3 after preoperative treaments were excluded because discriminating retrospectively between response to preoperative treatments or preoperative staging inaccuracy may be difficult.On the contrary, clinical T4a/b that were categorized as ypT4a/b after pathological examination of the surgical specimens were included (see Materials and Methods section, subheading 2.1 Patient population, page 3, lines 90-93.
Why did you include colorectal tumours complicating Crohn's disease, given that their clinical history is different?
AUTHORS’ RESPONSE: we agree with the reviewer that our manuscript is misleading regarding this point. Actually, the histopathological variable analyzed for its potential correlation with synchronous or metachronous peritoneal metastases is the presence vs. absence of Crohn's-like lymphoid reaction (CLR), not the diagnosis of Crohn disease. The Materials and Methods section, subheading 2.3 Pathological assessment, page 3, lines, Table 1 and Supplementary table were revised accordingly
The authors state that a distal margin of more than 2 cm must be respected in cancers of the upper rectum. To my knowledge, the exeresis must extend beyond 4 cm because of the risk of tumour islands in the mesorectum, which are present in up to 80% of cases.
AUTHORS’ RESPONSE: With the introduction of the TME surgery and preoperative chemoradiotherapy the 5 cm-rule has been abandoned and a distal resection margin ≥2 cm was found to be oncological sufficient [Park IJ et al. J. Gastrointest. Surg., 14 (2010), 1331]. A non-inferior oncological outcome of a distal resection margin of less than 1 cm has been described in patients receiving preoperative chemoradiotherapy for low rectal cancer [Rutkowski et al. Ann. Surg. Oncol., 15 (2008), 3124]. In patients not receiving preoperative chemoradiotherapy a distal resection margin of at least 2 cm seemed to be advisable in the past. New data of single center experiences reported [Nash GM et al. Dis. Colon Rectum, 53 (2010), 1365] a tumor free resection margin of less than 1 cm to be adequate to achieve excellent oncological results.
what classification was used to define post-operative morbidity?
AUTHORS’ RESPONSE: postoperative complications occurring within 30 days were scored according to the National Cancer Institute Common Terminology Criteria for Adverse Events, version 4.0 (see Materials and Methods section, subheading 2.2 Operative treatment, page 3, lines 11-112 and reference n. 18). Major morbidity was defined as postoperative complications requiring interventional radiology and/or endoscopy (grade 3), return to the operation room or to the Intensive Care Unit. Postoperative death represent grade 5.
what time frame was used to define the metachronous nature of peritoneal metastases?
AUTHORS’ RESPONSE: we considered as synchronous all metastases diagnosed before or at primary resection. All metastases diagnosed after primary resection were defined as metachronous. Different definitions of synchronous vs. metchronous metastases are available in the literature. We adopted a definition of “syncronously vs. metachronously detected metastases” because all of our patients underwent surgery for primary tumor resection and the abdominal cavity was explored carefully.
Of the 73 patients with synchronous peritoneal metastases, why was neoadjuvant chemotherapy only given to 23 of them?
AUTHORS’ RESPONSE: as reported in the Results section, subheading 3.1 Synchronous peritoneal metastases, page 6, lines 198-199, peritoneal metastases were detected intraoperatively in 30 patients. Seventeen patients underwent upfront surgery because of emergency presentation or other symptoms contraindicating preoperative systemic chemotherapy, such as bleeding or obstruction.
What was the post-operative morbidity? Was it assessed in the statistical analysis of risk factors for synchronous peritoneal metastases?
AUTHORS’ RESPONSE: as stated in the Results section, page 5, lines 177-179, severe (grades 3-5) postoperative complications occurred in 69 patients (19.6%). We did not assess the impact of postoperative complications on the occurrence of synchronous peritoneal metastasis, due to chronological reasons (i. e. an event (postoperative complications) cannot be the cause of something happening before (synchronous metastases). We also did not analyzed if postoperative complication rate was higher in patients with synchoronous peritoneal metastases because it is beyond the scope of the present study that focuses on the risk factors for peritoneal metastases. If the reviewer might be interested on these data, severe (grade 3-5) complications occurred in 11 of 73 patients with synchronous peritoneal metastases (13.9%) and 57 of 279 patients without synchronous metastases (20.4%). The difference was not statistically significant (p=0.322)
Finally, in the discussion, I will look at the role of neo-adjuvant treatment in T4
AUTHORS’ RESPONSE: We thank the reviewer for the very helpful suggestion. We modified the first paragraph of the Discussion section to clarify that the paradigm of T4 colorectal cancer management is continuously evolving, since the FOxTROT trial has demonstrate that a six-week preoperative systemic chemotherapy is associated with better 2-year disease control, that is a composite end-point including no resection, or macrosocopic incomplete resection after surgery (ie, residual tumor or metastases) or recurrence within 2 years. Nevertheless, we believe that it is worth to mention that both adiuvant and neoadjuvant systemic chemotherapy have never proven to specifically control the outgrowth of microscopic peritoneal spread into overt peritoneal metastases. This is an unmet clinical need that underscores the necessity of developing tailored strategies to prevent or diagnose and treat peritoneal metastases at an early stage.
Round 2
Reviewer 1 Report
Comments and Suggestions for Authors
The authors well response to the requests of R1 review.